# Irinotecan- vs. Oxaliplatin-Based Doublets in KRAS^G12C^-Mutated Metastatic Colorectal Cancer—A Multicentre Propensity-Score-Matched Retrospective Analysis

**DOI:** 10.3390/cancers15113064

**Published:** 2023-06-05

**Authors:** Vincenzo Formica, Cristina Morelli, Veronica Conca, Maria Alessandra Calegari, Jessica Lucchetti, Emanuela Dell’Aquila, Marta Schirripa, Marco Messina, Lisa Salvatore, Federica Lo Prinzi, Giovanni Dima, Giovanni Trovato, Silvia Riondino, Mario Roselli, Ferdinandos Skoulidis, Hendrik-Tobias Arkenau, Chiara Cremolini

**Affiliations:** 1Medical Oncology Unit, Department of Medicine of the Systems, University of Rome Tor Vergata, 00133 Rome, Italy; cristina.morelli@ptvonline.it (C.M.); silvia.riondino@uniroma2.it (S.R.); mario.roselli@ptvonline.it (M.R.); 2Department of Translational Research and New Technologies in Medicine and Surgery, University of Pisa, 56126 Pisa, Italy; v.conca@studenti.unipi.it (V.C.); dima_giovanni@libero.it (G.D.); chiara.cremolini@unipi.it (C.C.); 3Oncologia Medica, Comprehensive Cancer Center, Fondazione Policlinico Universitario Agostino Gemelli IRCCS, 00168 Rome, Italy; mariaalessandra.calegari@guest.policlinicogemelli.it (M.A.C.); lisa.salvatore@policlinicogemelli.it (L.S.); giovanni.trovato01@icatt.it (G.T.); 4Fondazione Policlinico Universitario Campus Bio-Medico, Via Alvaro del Portillo, 200, 00128 Rome, Italy; j.lucchetti@policlinicocampus.it (J.L.); federica.loprinzi@unicampus.it (F.L.P.); 5Regina Elena National Cancer Institute, Medical Oncology 1, 00144 Rome, Italy; emanuela.dellaquila@ifo.it; 6Department of Oncology and Hematology, Belcolle Hospital, Azienda Sanitaria Locale Viterbo, 01100 Viterbo, Italy; marta.schirripa@asl.vt.it; 7UOC Oncologia Medica, ARNAS Ospedali Civico Di Cristina Benefratelli, 90127 Palermo, Italy; marco.messina@arnascivico.it; 8Oncologia Medica, Università Cattolica del Sacro Cuore, 00186 Rome, Italy; 9Department of Thoracic-Head & Neck Medical Oncology, Division of Cancer Medicine, The University of Texas MD Anderson Cancer Center, Houston, TX 77030, USA; fskoulidis@mdanderson.org; 10Ellipses Pharma, London W1J 8LG, UK; tobi@ellipses.life

**Keywords:** KRAS^G12C^ mutation, metastatic colorectal cancer, irinotecan, oxaliplatin

## Abstract

**Simple Summary:**

The sensitivity to chemotherapy of KRASG12C-mutated colorectal cancer has been investigated to verify whether the combination of chemotherapy plus a KRASG12C-inhibitor might become the standard of care in the near future. To this aim, the present retrospective study was designed to assess the performance of irinotecan vs. oxaliplatin in the first-line treatment of KRASG12C-mutated mCRC patients and provide support for first-line decision making. In this setting of patients treated with FOLFIRI or FOLFOX +/− bevacizumab, irinotecan and oxaliplatin were compared using a propensity-score-matched analysis. The survival superiority of irinotecan was demonstrated over oxaliplatin in KRASG12C-mutated patients, while no differences were observed in a control cohort of KRASG12D-mutated patients. This should be considered when investigating chemotherapy plus targeted agent combinations.

**Abstract:**

Background: KRAS^G12C^-mutated metastatic colorectal cancer (mCRC) has recently been recognized as a distinct druggable molecular entity; however, there are limited data on its sensitivity to standard chemotherapy. In the near future, the combination of chemotherapy plus a KRAS^G12C^-inhibitor might become the standard of care; however, the optimal chemotherapy backbone is unknown. Methods: A multicentre retrospective analysis was conducted including KRAS^G12C^-mutated mCRC patients treated with first-line FOLFIRI or FOLFOX +/− bevacizumab. Both unmatched and propensity-score-matched analysis (PSMA) were conducted, with PSMA controlling for: previous adjuvant chemotherapy, ECOG PS, use of bevacizumab in first line, timing of metastasis appearance, time from diagnosis to first-line start, number of metastatic sites, presence of mucinous component, gender, and age. Subgroup analyses were also performed to investigate subgroup treatment–effect interactions. KRAS^G12D^-mutated patients were analysed as control. Results: One hundred and four patients treated with irinotecan-(N = 47) or oxaliplatin-based (N = 57) chemotherapy were included. In the unmatched population, objective response rate (ORR) and median (m) progression-free and overall survival (mPFS and mOS) were comparable between the treatment arms. However, a late (>12 months) PFS advantage was observed with irinotecan (HR 0.62, *p* = 0.02). In the PSMA-derived cohort, a significant improvement with irinotecan vs. oxaliplatin was observed for both PFS and OS: 12- and 24-month PFS rates of 55% vs. 31% and 40% vs. 0% (HR 0.40, *p* = 0.01) and mOS 37.9 vs. 21.7 months (HR 0.45, *p* = 0.045), respectively. According to the subgroup analysis, interaction effects between the presence of lung metastases and treatment groups were found in terms of PFS (*p* for interaction = 0.08) and OS (*p* for interaction = 0.03), with a higher benefit from irinotecan in patients without lung metastases. No difference between treatment groups was observed in the KRAS^G12D^-mutated cohort (N = 153). Conclusions: First-line irinotecan-based regimens provided better survival results in KRAS^G12C^-mutated mCRC patients and should be preferred over oxaliplatin. These findings should also be considered when investigating chemotherapy plus targeted agent combinations.

## 1. Introduction

Colorectal cancer (CRC) represents the third most common cancer type and the fourth leading cause of cancer-related death in Western countries [1]. In about half of cases metastatic disease is observed, either at diagnosis or as metachronous occurrence [2]. Median overall survival (mOS) for metastatic CRC (mCRC) is currently 18–24 months in the unselected population, but it varies according to the *RAS* and *BRAF* mutational status and primary tumour side [3]. *RAS*-mutated tumours are associated with a worse prognosis than *RAS* and *BRAF* wild-type tumours [4], which is also due to intrinsic resistance to the anti-EGFR agents cetuximab and panitumumab [5]. Activating *KRAS* mutations are frequent across many solid tumour types, and in CRC they are present in approximately 40% of cases [6]. They have been considered undruggable for a long time due to their unstable inhibition by currently available drugs available.

More recently, stable inhibitors of KRAS^G12C^-mutated protein have been developed and have demonstrated clinical benefit in phase I/II studies, as well as in combination with cetuximab [7], while phase III trials testing these agents against the standard of care in second or later lines of treatment are currently ongoing [8].

Nonetheless, today, the standard first-line therapy of KRAS^G12C^-mutated mCRC patients remains irinotecan- or oxaliplatin-based doublets, with few data exploring possible differences between the two drugs in this specific molecular subtype. The aim of the present study was to retrospectively review the performance of irinotecan vs. oxaliplatin in the first-line treatment of KRAS^G12C^-mutated mCRC patients and provide support for first-line decision making.

## 2. Patients and Methods

A multicentre retrospective analysis was conducted including patients from 7 Italian institutions with histologically confirmed CRC and measurable metastatic disease treated with a first-line chemotherapy. Only KRAS^G12C^-mutated patients treated between 2009 and 2021 with either FOLFIRI +/− bevacizumab or FOLFOX +/− bevacizumab were selected. KRAS mutational status was determined using either real-time PCR, pyrosequencing, or next-generation sequencing according to local procedure. Patients with microsatellite instability high/mismatch repair deficient tumours and BRAF-mutated tumours were not included. For the present analysis, the cut-off date for survival follow-up of included patients was 25 March 2022. All patients signed informed consent for collection of anonymized data for retrospective analyses at the time of the first-line therapy. The research was carried out in accordance with The Code of Ethics of the World Medical Association (Declaration of Helsinki).

Outcome measures were progression-free survival (PFS), defined as the time from first-line start until radiological or clinical disease progression or death; OS, defined as the time from first-line start until death from any cause; and radiological objective response rate (ORR) according to RECIST 1.1 criteria.

Survival curves were drawn according to the Kaplan–Meier method and compared with the log-rank test. Accompanying hazard ratios (HR) using a univariate Cox-proportional hazards model were estimated. Chi-square test was used for difference in categorical variables.

Difference between irinotecan and oxaliplatin was firstly assessed in the unmatched KRAS^G12C^-mutated population. Subsequently, to overcome possible selection bias in choosing first-line irinotecan vs. oxaliplatin, a propensity score and exact matching (PSM-EM) analysis was performed, thus ensuring balanced key baseline characteristics between the two treatment groups. Propensity score was estimated upon the following possibly influencing variables: previous oxaliplatin-based adjuvant chemotherapy (yes vs. no), ECOG PS (0 vs. 1 or more), use of bevacizumab in combination with first-line chemotherapy (yes vs. no), timing of metastasis appearance (synchronous vs. metachronous), time elapsed from histological diagnosis to first-line commencement (< vs. >12 months), number of metastatic sites (1 vs. 2 or more), and presence of mucinous component (yes vs. no). Exact matching was performed by gender (male vs. female) and age (< vs. >65 years).

Pairs of irinotecan- and oxaliplatin-treated patients were matched on the logit of the propensity score using the near neighbour method and a calliper of width equal to 0.2 of the standard deviation of the logit of the propensity score.

Subgroup analyses for key outcome measures were also performed to investigate significant treatment effect/subgroup interaction and presented as forest plot with interaction *p* values.

Treatment activity was also investigated in a control population of patients harbouring the KRAS^G12D^ mutation exactly matched for gender and age and selected from the databases of the 7 participating centres in order to validate the KRAS^G12C^-specificity of the primary findings.

A statistically significant finding was defined as a two-sided *p* value < 0.05. All analyses were carried out with R (version 4.0.3).

## 3. Results

Out of 4843 screened patients, 122 had KRAS^G12C^-mutated (2.52%) mCRC, and 104 of them met the inclusion criteria and were treated with first-line FOLFIRI (12 patients), FOLFIRI plus bevacizumab (35 patients), FOLFOX (14 patients), or FOLFOX plus bevacizumab (43 patients). Patients’ characteristics are reported in Table 1. As of March 2022, 96 (92%) patients had reached the progression endpoint and 78 (75%) the survival endpoint. Median follow-up of surviving patients was 20.7 months (range 7.8 to 118.2 months). Second-line treatment was received by 52% of patients, in most cases including the cytotoxic agent not received in the first line. No patient received the KRAS-specific inhibitor.

In the whole cohort of 104 patients, ORR, mPFS, and mOS were 58%, 10.2 months (95% CI 9.1–11.6 months), and 24.9 months (95% CI 20.2–29.5 months), respectively.

In the unmatched analysis, ORR was 57% vs. 58% for irinotecan- vs. oxaliplatin-treated patients, respectively.

Furthermore, mPFS was comparable between irinotecan- vs. oxaliplatin-treated patients: 10.5 vs. 9.9 months, respectively. However, at approximately 12 months, the curves separated in favour of irinotecan with a significant improvement in 12-, 18- and 24-month PFS rates were 38% vs. 30%, 32% vs. 13%, and 22% vs. 4%, for irinotecan vs. oxaliplatin, respectively; HR 0.62 (95% CI 0.41–0.94) and *p* = 0.02 (Figure 1A).

A numerically higher mOS was observed for irinotecan as compared to oxaliplatin, with 28.9 vs. 21.1 months, respectively; HR 0.70 (95% CI 0.44–1.10) and *p* = 0.12 (Figure 1B). 

To account for the imbalance in possible influencing factors, a PSM-EM analysis was performed. Background variables that were taken into account for patients treated with FOLFIRI +/− bevacizumab or FOLFOX +/− bevacizumab were the following: previous oxaliplatin-based adjuvant chemotherapy (yes vs. no), ECOG PS (0 vs. 1 or more), use of bevacizumab in combination with first-line chemotherapy (yes vs. no), timing of metastasis appearance (synchronous vs. metachronous), time elapsed from histological diagnosis to first-line commencement (< vs. >12 months), number of metastatic sites (1 vs. 2 or more), presence of mucinous component (yes vs. no), gender (male vs. female), and age (< vs. >65 years).

In total, 44 patients (22 patients per arm) were found to adequately match and were selected for the matched analysis. The advantage of irinotecan was even more pronounced in the PSM-EM-derived cohort, with 12-, 18- and 24-month PFS rates of 55% vs. 31%, 50% vs. 14%, and 40% vs. 0% for irinotecan vs. oxaliplatin, respectively; HR 0.40 (95% CI 0.20–0.83) and *p* = 0.01 (Figure 2A). The mOS was 37.9 vs. 21.7 months for irinotecan vs. oxaliplatin, respectively; HR 0.45 (95% CI 0.21–0.98) and *p* = 0.045 (Figure 2B). ORR was 73% vs. 68%, respectively.

In order to investigate whether the survival advantage of irinotecan was driven by specific clinical features, subgroup analyses were performed. Among all the tested subgroups, interaction effects between the presence or not of lung metastases and treatment groups were found in terms of PFS (*p* for interaction = 0.08) and OS (*p* for interaction = 0.03) (Figure 3). By analysing PFS and OS separately in the two subgroups of patients without and with lung metastases, the survival advantage of irinotecan was significant for both PFS and OS only in patients without lung metastases (62 patients): mPFS 11.3 vs. 9.1 months (HR 0.37 (95% CI 0.21–0.66), *p* = 0.0007) and mOS 27.2 vs. 17.3 months (HR 0.43 (95% CI 0.24–0.78), *p* = 0.005) for irinotecan and oxaliplatin, respectively (Figure 4A,B). ORR was numerically superior with irinotecan in this subgroup: 69% vs. 47%, with a chi-square *p* value 0.087. No survival difference by treatment was observed in patients with lung metastases, with *p* = 0.85 and 0.55 for PFS and OS, respectively (Figure 4C,D).

A population of 153 KRAS^G12D^-mutated mCRC patients treated with either FOLFIRI +/− bevacizumab (61 patients) or FOLFOX +/− bevacizumab (92 patients) in the first-line setting, exactly matched for age and gender with the KRAS^G12C^-mutated cohort, was used as control group. In KRAS^G12D^-mutated patients, no difference was observed between irinotecan- and oxaliplatin-based regimens in terms of both PFS (*p* = 0.36) and OS (*p* = 0.29) (Figure 5A,B).

## 4. Discussion

In the present retrospective multicentre study including 104 KRAS^G12C^-mutated mCRC patients treated with either FOLFIRI or FOLFOX +/− bevacizumab in the first-line setting, we were able to demonstrate a long-term survival benefit of irinotecan-based regimens in this specific molecular subtype. This effect was maintained and even reinforced after adjusting for possible imbalances with a propensity-score-matched analysis based on the receipt of adjuvant oxaliplatin, ECOG PS, bevacizumab within the first-line regimen, timing of metastasis appearance, time elapsed from histological diagnosis to first-line commencement, number of metastatic sites, presence of mucinous component, gender, and age.

The benefit of irinotecan was especially evident in patients without lung metastases, with a mOS > 27 months when KRAS^G12C^-mutated patients were treated with an irinotecan-based first-line chemotherapy.

The superiority of irinotecan-based regimens could not be demonstrated in the control cohort of 153 patients harbouring the KRAS^G12D^ mutation.

The choice of a KRAS^G12D^-mutated cohort as control was essentially based on the effort to make the control group as most homogeneous as possible and on the high prevalence of this alteration in CRC. Moreover, G12D-specific inhibitors are already under investigation in phase I trials (e.g., NCT05382559); therefore, collecting more information about patients with tumours harbouring this specific KRAS mutation is of interest.

This is not the first time that irinotecan was shown to be superior to oxaliplatin in *KRAS*-mutated mCRC. In a phase II randomized trial of the AIO cooperative group, where first-line capecitabine/irinotecan/bevacizumab was compared to capecitabine/oxaliplatin/bevacizumab in unselected patients, the irinotecan-based regimen showed superiority for OS in the *KRAS* codon 12–13-mutated subgroup (74 patients): mOS 28.7 vs. 18.8 months, respectively, *p* = 0.03 [9]. Analyses by sub-mutations in general, and for the KRAS^G12C^ mutation in particular, were not performed in this study.

In another retrospective series including 128 KRAS-mutated patients, a survival advantage with irinotecan-based first-line chemotherapy was also reported in codon 12 mutant patients (mOS 42.7 months) [10].

In a recent study by Ciardiello et al. including KRAS^G12C^-mutated mCRC patients, a numerically higher mOS was observed in irinotecan-treated (29 patients) vs. oxaliplatin-treated patients (66 patients), at 22 vs. 18 months, respectively [11]. In this study, a superiority of the triplet regimen FOLFOXIRI administered to 16 patients was observed; however, no PSM analysis was performed. In our study, patients treated with FOLFOXIRI were excluded since the analysis was mainly focused on the difference between oxaliplatin and irinotecan in consideration of the fact that, in common clinical practice, most of patients are deemed candidates for a doublet regimen of either FOLFIRI or FOLFOX. The subgroup analyses of randomized trials of FOLFOXIRI versus FOLFIRI (TRIBE) or FOLFOX (TRIBE2) according to KRAS^G12C^ mutation would provide useful information about the added value of the intensified chemotherapy regimen in this patients’ subgroup.

A signal of different sensitivity to anticancer therapy based on the metastatic pattern was observed in our analysis. This has recently become a focus of research also in trials with immunotherapy, where lung and liver metastases from CRC seem to show distinct genomic profiles and immune microenvironments, which would explain a differential response to the treatment [12,13,14,15].

Some mutations, such as those affecting *TCF7L2*, *APC*, *FLT3*, *TP53*, and *ZFP36L2*, seem to preferentially occur in lung metastases with either a polyphyletic or monophyletic mechanism [16]. It would be of interest to specifically assess whether these mutations might interact with irinotecan or oxaliplatin sensitivity.

Moreover, a more in-depth assessment of baseline CT scans would be useful to work out the influence of disease burden on chemotherapy sensitivity. CT scans were not available for the analysis in our study; however, we used the total number of metastatic sites (‘2 or more’ vs. ‘1’) as a “proxy” of disease burden (Figure 3).

It is well known that rectal cancer more frequently spreads to lungs than colon cancer. In our cohort, only 10 patients had a rectal primary tumour, with half (n = 5) of them presenting with lung metastases. Because of the limited number of patients, no further stratified analyses were specifically conducted for rectal cancer patients.

Differences in the sites of metastatic spread according to KRAS mutational status were previously reported. In a study by Tie et al. [17], a higher prevalence of KRAS mutations in lung versus liver metastases was found (62% vs. 32%, *p* = 0.003), and a higher risk of lung relapse was evidenced among those with KRAS-mutated primary tumours (HR = 2.1, *p* = 0.007).

Intra-tumour heterogeneity between primary tumour and lung metastases has also been confirmed in KRAS^G12C^-mutated patients [18]. However, whether this may influence sensitivity to irinotecan, or the interaction between drug pharmacodynamics and host microenvironment, is yet to be assessed [19,20]. The observed difference in chemotherapy sensitivity in KRAS^G12C^-mutated patients is perhaps not due to the KRAS^G12C^ mutation itself but rather to co-occurring alterations that are possibly more frequent in KRAS^G12C^-mutated tumours. Following this hypothesis, the study of pathways involving the topoisomerase I enzyme, the target of the active form of irinotecan SN38, would be recommended.

The evidence of interaction between treatment effect and pattern of metastases in KRAS^G12C^-mutated mCRC is completely unprecedented and should be cautiously interpreted given that the present study is retrospective, and the subgroup analysis was unplanned. However, it would deserve further investigation.

## 5. Conclusions

In conclusion, our data suggest that, currently, in the absence of approved targeted agents in the first-line setting, when a doublet chemotherapy is chosen, irinotecan-based regimens should be preferred, especially in patients without lung metastases. Moreover, possible synergic effects should be taken into consideration if combinations of ‘chemo-targeted’ therapies with novel KRAS^G12C^-inhibitors are tested in clinical trials, since these new drugs might either potentiate the effect of less efficacious regimens, such as oxaliplatin-based doublets, or further enhance the beneficial effect of more efficacious regimens, such as those that are irinotecan-based [21].

## Figures and Tables

**Figure 1 cancers-15-03064-f001:**
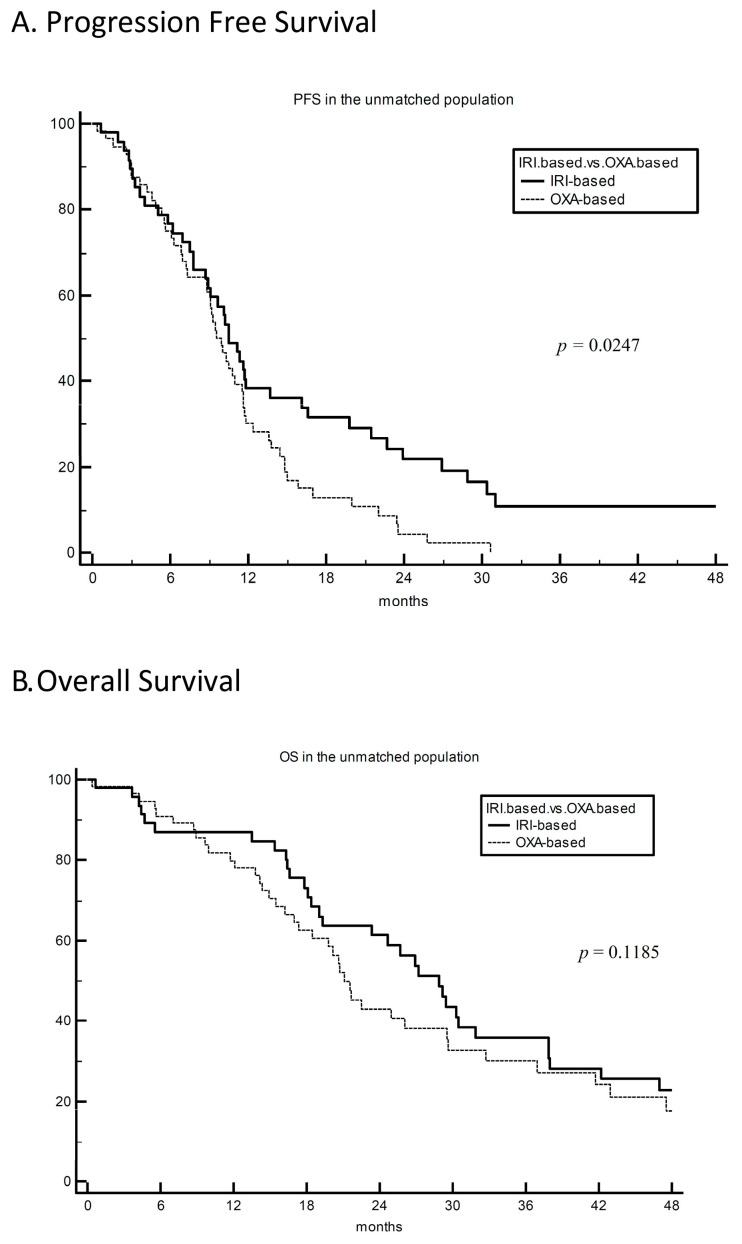
PFS (panel (**A**)) and OS (panel (**B**)) analysis for first-line irinotecan-based (FOLFIRI +/− bevacizumab) vs. oxaliplatin-based (FOLFOX +/− bevacizumab) chemotherapy doublet in the unmatched primary population of 104 KRAS^G12C^-mutated metastatic colorectal cancer patients. IRI-based: irinotecan-based. OXA-based: oxaliplatin-based. PFS: progression-free survival. OS: overall survival.

**Figure 2 cancers-15-03064-f002:**
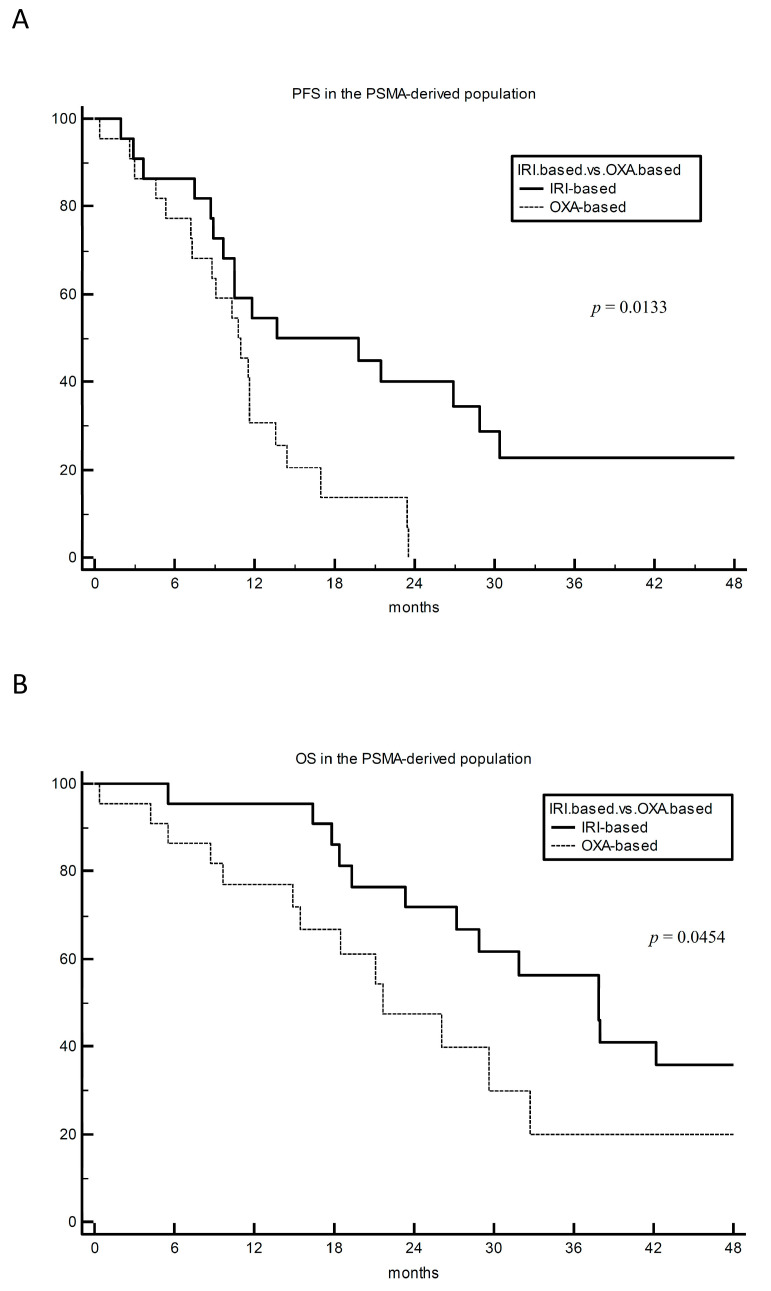
PFS (panel (**A**)) and OS (panel (**B**)) analysis for first-line irinotecan-based (FOLFIRI +/− bevacizumab) vs. oxaliplatin-based (FOLFOX +/− bevacizumab) chemotherapy doublet in the propensity score matching analysis (PSMA)-derived population of 44 KRAS^G12C^-mutated metastatic colorectal cancer patients. Propensity score was based on receipt of adjuvant oxaliplatin, ECOG PS, receipt of bevacizumab within the first-line regimen, timing of metastasis appearance, time elapsed from histological diagnosis to first-line commencement, number of metastatic sites, presence of mucinous component, gender, and age. IRI-based: irinotecan-based. OXA-based: oxaliplatin-based. PFS: progression-free survival. OS: overall survival.

**Figure 3 cancers-15-03064-f003:**
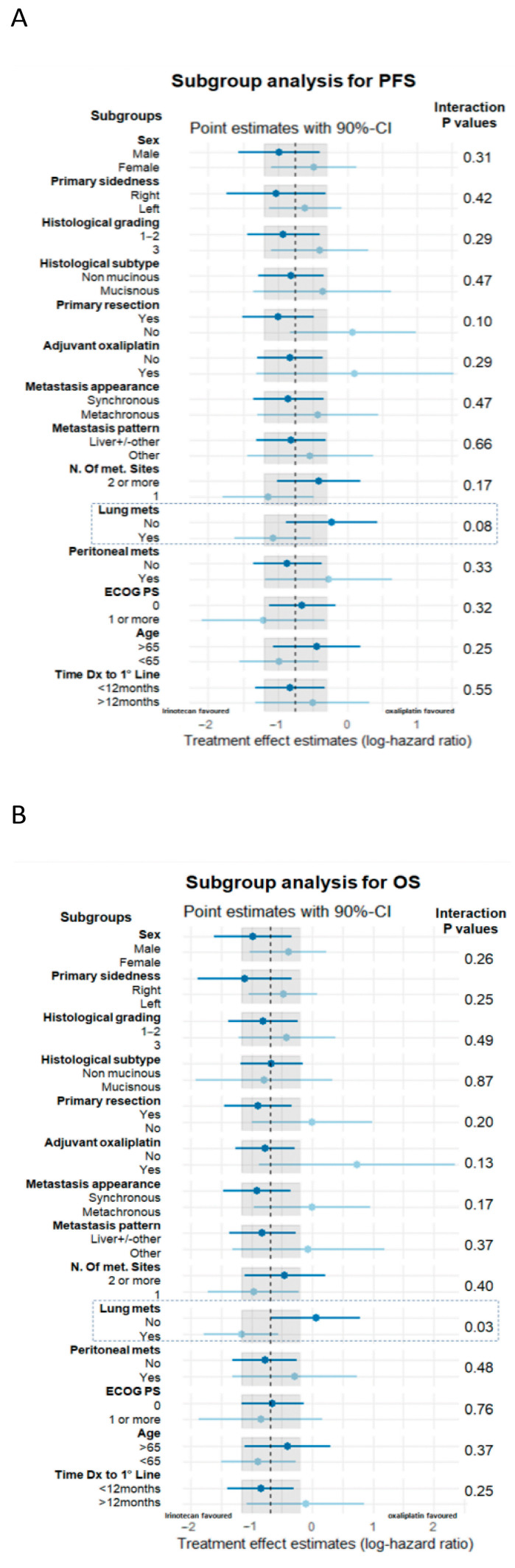
Subgroup analysis for treatment effect–key variables interaction for PFS (**A**) and OS (**B**). The dashed line and grey areas indicate the log-hazard ratio and 90% confidence intervals in the whole population of 104 patients. PFS: progression-free survival. OS: overall survival. CI: confidence interval. Dx: diagnosis.

**Figure 4 cancers-15-03064-f004:**
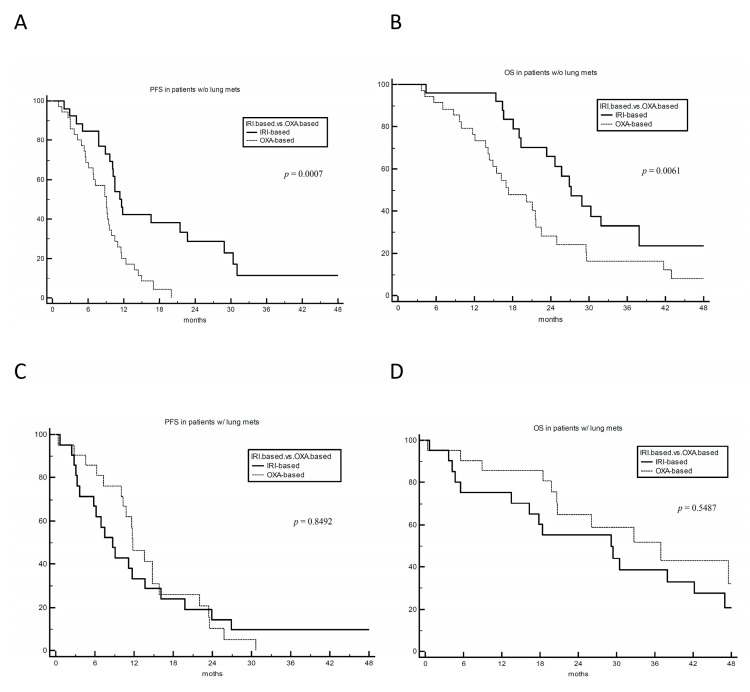
PFS (panels (**A**,**C**)) and OS (panels (**B**,**D**)) by treatment and absence (panels (**A**,**B**)) or presence (panels (**C**,**D**)) of lung metastasis in the primary population of 104 KRAS^G12C^-mutated metastatic colorectal cancer patients. w/o: without. w/: with. IRI-based: irinotecan-based chemotherapy. OXA-based: oxaliplatin-based chemotherapy. PFS: progression-free survival. OS: overall survival.

**Figure 5 cancers-15-03064-f005:**
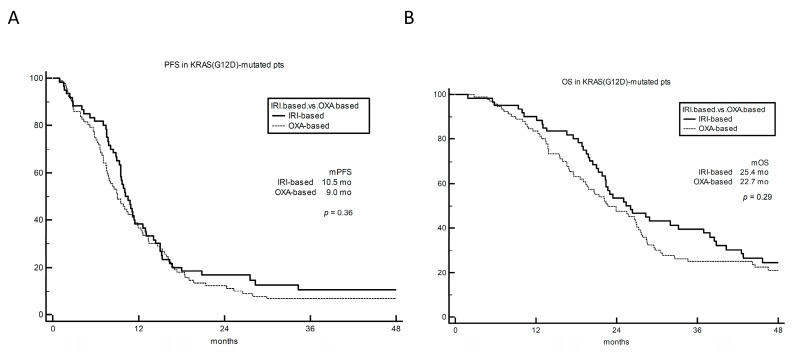
PFS (panel (**A**)) and OS (panel (**B**)) by treatment group in 153 patients with KRAS^G12D^-mutated mCRC treated with first-line chemotherapy. OXA-based: oxaliplatin-based chemotherapy. PFS: progression-free survival. OS: overall survival.

**Table 1 cancers-15-03064-t001:** Characteristics of the 104 KRAS^G12C^-mutated metastatic colorectal cancer patients included in the study.

Characteristics	% (N = 104)
**Sex**	
Male	40% (42)
Female	60% (62)
**Age**	
<65	54% (56)
≥65	46% (48)
**Primary tumour sidedness**	
Right	37% (38)
Left	63% (66)
**Mucinous histology**	
Yes	14% (15)
No	86% (89)
**Metastasis at diagnosis**	
Yes	69% (72)
No	31% (32)
**Number of metastatic sites**	
1	41% (43)
≥2	59% (61)
**Metastatic sites**	
Liver	77% (80)
Lung	41% (43)
Peritoneum	22% (23)
**First-line treatment**	
Oxaliplatin-based doublet	55% (57)
Irinotecan-based doublet	45% (47)
**Bevacizumab use in first line**	
Yes	75% (78)
No	25% (26)

## Data Availability

All raw anonymized data are available upon request via email to the corresponding author.

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
