# Peer review of "Irinotecan- vs. Oxaliplatin-Based Doublets in KRASG12C-Mutated Metastatic Colorectal Cancer—A Multicentre Propensity-Score-Matched Retrospective Analysis"

_cancers, 2023, doi:10.3390/cancers15113064_

Round 1

Reviewer 1 Report

The study by Formica et al entitled "Irinotecan- vs oxaliplatin-based doublets in KRASG12C-mutated metastatic colorectal cancer. A multicenter propensity-matched retrospective study"  investigated the optimal chemotherapy backbone in patients with KRASG12C mutated metastatic colorectal cancer (mCRC). This is important for optimizing treatment strategies in patients with mCRC. Additionally, it is increasingly important with the upcoming KRAS inhibitors to interpret results of upcoming phase III clinical trials comparing the efficacy of KRAS inhibitors plus anti-EGFR with standard chemotherapy. Additionally it can be important for future studies intending to combine KRAS inhibitors with a chemotherapy backbone. 

This study is well written and presented in a clear way. However there are some issues that need to be addressed before publication.

1.      Outcomes are compared to the outcomes of patients with a KRASG12D mutation in the tumor. This is the most frequently occurring KRAS mutation in colorectal cancer, and some studies show a more favorable prognosis of this mutation compared to KRASG12C mutation. Rationale for comparing the current outcomes to this specific mutation is lacking and how are results compared to other KRAS mutations such as KRASG12V? 

2.       The authors separate left and right sidedness but rectal cancer is not defined as a separate entity, while rectal cancer is different to some extent. For instance lung metastases are generally more frequently observed in patients with rectal cancer and prognosis can be different. This study shows a difference in survival with oxaliplatin versus irinotecan in patient with and without lungmetastases. Rectal cancer should be added and analyzed separately.

3.       The authors do not mention the mismatch repair status of the tumors while it is known that MMR deficient and proficient cancers have a different response to the treatment and also differ in survival. The MMR status should be added in the study results and analyses.

4.       The authors describe that they excluded patients treated with triple therapy since this study aimed at comparing differences in treatment response of oxaliplatin versus irinotecan in this particular subgroup of KRAS mutation. Additionally the authors state that in clinical practice most patients are candidates for oxaliplatin or irinotecan doublet therapy. With the increasing literature, including their own, showing that triple therapy can improve survival compared to doublet treatment it is specifically of clinical relevance to also compare the presented study results to matched data of patients who have been treated with folfoxiri. This is also of importance to the field and I would suggest adding this data to the manuscript. Additionally, it is important for future studies investigating novel treatment strategies for this specific subgroup comparing outcome to current standard of care. Current standard of care increasingly comprises triple therapy, especially in younger patients in good clinical condition.

Author Response

1)Outcomes are compared to the outcomes of patients with a KRASG12D mutation in the tumor. This is the most frequently occurring KRAS mutation in colorectal cancer, and some studies show a more favorable prognosis of this mutation compared to KRASG12C mutation. Rationale for comparing the current outcomes to this specific mutation is lacking and how are results compared to other KRAS mutations such as KRASG12V? 

RE:  We thank the reviewer for the opportunity to clarify this point. We believed that a cohort of KRAS non-G12C mutated mCRC patients serving as control group would have made our results more robust, somehow supporting the idea that they could be actually dependent on the G12C mutation and not applicable to all KRAS mutations. In an effort to make the control group as homogeneous as possible we chose to focus on a single KRAS mutation and we selected the G12D alteration because it is the most common KRAS mutation in CRC and G12D-specific inhibitors are already under investigation in phase I trials. We think that collecting more information about patients with tumours harbouring the KRASG12D mutation is of interest in a future perspective. We have now added a comment in the discussion section accordingly.

2)The authors separate left and right sidedness but rectal cancer is not defined as a separate entity, while rectal cancer is different to some extent. For instance lung metastases are generally more frequently observed in patients with rectal cancer and prognosis can be different. This study shows a difference in survival with oxaliplatin versus irinotecan in patient with and without lung metastases. Rectal cancer should be added and analyzed separately.

RE: we thank the reviewer for the suggestion. Unfortunately only 10 patients out of 104 had a rectal primary tumour, thus not allowing to perform additional informative sub-analyses. The issue of the link between rectal cancer and lung metastases is now addressed in the discussion section.

3)The authors do not mention the mismatch repair status of the tumors while it is known that MMR deficient and proficient cancers have a different response to the treatment and also differ in survival. The MMR status should be added in the study results and analyses.

RE: we thank the reviewer for this suggestion. Patients with MMR deficient tumors were actually excluded, this is now well specified in the methods section

4)The authors describe that they excluded patients treated with triple therapy since this study aimed at comparing differences in treatment response of oxaliplatin versus irinotecan in this particular subgroup of KRAS mutation. Additionally the authors state that in clinical practice most patients are candidates for oxaliplatin or irinotecan doublet therapy. With the increasing literature, including their own, showing that triple therapy can improve survival compared to doublet treatment it is specifically of clinical relevance to also compare the presented study results to matched data of patients who have been treated with folfoxiri. This is also of importance to the field and I would suggest adding this data to the manuscript. Additionally, it is important for future studies investigating novel treatment strategies for this specific subgroup comparing outcome to current standard of care. Current standard of care increasingly comprises triple therapy, especially in younger patients in good clinical condition.

RE: we thank the reviewer for the suggestion. We do recognize that the performance of the FOLFOXIRI regimen in KRASG12C-mutated tumors was not the aim of this work, although it would be of extreme interest, given the possible superiority of this regimen as compared to the doublet chemotherapy especially in fit patients not treated with adjuvant oxaliplatin. However, we felt that chemotherapy doublets will be the chemotherapy backbone in most cases and trials investigating the efficacy of KRASG12C inhibitor plus chemotherapy combination would most likely adopt doublets instead of a triplet regimen. The outcome with FOLFOXIRI in a real-life scenario has already been shown in the work by Ciardiello et al (Ciardiello D, ESMO Open, 2022). Nonetheless, a comment in the discussion section has been added to highlight the interest in assessing post-hoc the interaction between assigned treatment and KRAS submutation in the FOLFOXIRI phase III trials, such as TRIBE and TRIBE-2, where FOLFIRI (TRIBE) and FOLFOX (TRIBE-2) were used as control arm.

Reviewer 2 Report

KRAS G12C was very important biomarker of  colorectal cancer. This study is very impressive and important evidence.

1.Please show  back ground of propensity score matched population of both OX-based and IRI-based regimen.

2.Do you have any information of post-treatment?( rate, type of regimen, such as KRAS G12C targeted therapy)

3.The differences in biology in pulmonary metastasis cases are interesting, but I don't think the data are limited to KRAS G12C.

If there are any studies that have examined differences in biology between lung metastases and other organ metastases in KRAS mutations, please describe them.

Author Response

1)Please show  background of propensity score matched population of both OX-based and IRI-based regimen.

RE: background variables taken into account are now clearly reported in the results section.

2)Do you have any information of post-treatment?(rate, type of regimen, such as KRAS G12C targeted therapy)

RE: second line data are now reported in the results. Moreover it is now specified that no patient received a  KRAS G12C inhibitor

3) The differences in biology in pulmonary metastasis cases are interesting, but I don't think the data are limited to KRAS G12C. If there are any studies that have examined differences in biology between lung metastases and other organ metastases in KRAS mutations, please describe them.

RE: The description of a study by Tie J et al investigating difference in KRAS mutations between distinct metastatic sites has been added in the discussion.

Reviewer 3 Report

This propensity matched analysis assess the role of irinotecan and oxaliplatin in the treatment of mCRC patients carrying G12C/D KRAS mutations. The authors convincingly show following well described propensity matching the efficacy of irinotecan based regimens in KRAS G12C cancer. Clearly there are limitations from small sample size and retrospective nature. I have a few comments:

Why do authors feel there is an advantage in G12C patients alone?

Given Kras mutations in G12D would engage similar downstream mechanisms how can this be explained?

How are other mutations within the tumours controlled for?

Is there genuinely evidence that this is a Kras specific effect?

Commentary in the discussion on lung metastases is important given how different responses are - do authors feel other mutations drive this pattern of disease and are therefore less sensitive? Or does this simply reflect burden of disease.

How was burden of disease controlled for in the study? 

Occasional issues with grammar - should be checked

Author Response

1)Why do authors feel there is an advantage in G12C patients alone? Given Kras mutations in G12D would engage similar downstream mechanisms how can this be explained? How are other mutations within the tumours controlled for? Is there genuinely evidence that this is a Kras specific effect?

RE: We agree with the reviewer that probably the observed difference in chemotherapy sensitivity is not due to the KRAS G12C mutation itself but rather to co-occuring alterations. These co-occuring alterations are possibly more frequent in G12C than in non-G12C mutated tumors and would confer higher sensitivity to irinotecan. This hypothesis is now discussed in the discussion. No formal control for co-occurring mutations except from BRAF mutation and microsatellite instability has been conducted.

2) Commentary in the discussion on lung metastases is important given how different responses are - do authors feel other mutations drive this pattern of disease and are therefore less sensitive?

 RE: other possible mutations involved in lung metastases are described in the paper by Chen et al (ref 16). We have now added a comment on the importance of investigating their possible interaction with irinotecan or oxaliplatin efficacy in the discussion.

3) Or does this simply reflect burden of disease. How was burden of disease controlled for in the study?

The total number of metastatic sites (variable dichotomized in ‘2 or more’ vs ‘1’) has been used as a “proxy” of disease burden and is reported in figure 3. We do agree that a more accurate tumor burden quantification would have been useful, however we had no access to baseline CT scans. This is now discussed in the discussion

Round 2

Reviewer 1 Report

The authors have addressed and answered all points raised and adapted the discussion section. Although I still do think that the comparison with triple therapy is important to the field, I respect the arguments of the authors and the point has been raised in the discussion section of the manuscript. Therefore, the manuscript has been sufficiently revised for acceptance of publication in this journal. 

One small remark: (< vs >) is incorrect, it should be: (< or ≥) , or (≤ or >)

Reviewer 3 Report

I appreciate your responses and accept that there are not many changes that can be brought to the work in its current form.